# Clinical Validation and Post-Implementation Performance Monitoring of a Neural Network-Assisted Approach for Detecting Chronic Lymphocytic Leukemia Minimal Residual Disease by Flow Cytometry

**DOI:** 10.3390/cancers17101688

**Published:** 2025-05-17

**Authors:** Jansen N. Seheult, Gregory E. Otteson, Matthew J. Weybright, Michael M. Timm, Wenchao Han, Dragan Jevremovic, Pedro Horna, Horatiu Olteanu, Min Shi

**Affiliations:** 1Cell Kinetics Laboratory, DLMP, Mayo Clinic, Rochester, MN 55905, USA; otteson.gregory@mayo.edu (G.E.O.); weybright.matthew@mayo.edu (M.J.W.); timm.michael@mayo.edu (M.M.T.); jevremovic.dragan@mayo.edu (D.J.); horna.pedro@mayo.edu (P.H.); olteanu.horatiu@mayo.edu (H.O.); shi.min@mayo.edu (M.S.); 2Division of Computational Pathology and Informatics, DLMP, Mayo Clinic, Rochester, MN 55905, USA; han.wenchao@mayo.edu

**Keywords:** artificial intelligence, deep neural network, flow cytometry, chronic lymphocytic leukemia, minimal residual disease, clinical validation, performance monitoring

## Abstract

Flow cytometric detection of minimal/measurable residual disease (MRD) in chronic lymphocytic leukemia (CLL) is critical for prognostication and therapy guidance but remains time-consuming and subjective. In this study, we evaluated a deep neural network (DNN)-assisted human-in-the-loop approach for CLL MRD detection in a clinical laboratory setting. We performed comprehensive clinical validation with 240 samples, demonstrating 97.5% concordance with manual analysis (sensitivity 100%, specificity 95%) and excellent correlation for quantitative assessment (r = 0.99, Deming slope = 0.99) compared to the manual reference method. Precision studies confirmed high repeatability across multiple operators, and analytical sensitivity was verified at 0.002% MRD. Our post-implementation monitoring system identified 2.97% of cases with input data drift, primarily high-burden CLL and non-CLL neoplasms. The DNN-assisted workflow reduced average analysis time by 60.3% compared to manual methods. This study provides a model for responsible AI integration in clinical laboratories, balancing automation benefits with expert oversight while maintaining diagnostic accuracy.

## 1. Introduction

The implementation of artificial intelligence (AI) and machine learning (ML) in clinical laboratory settings has demonstrated significant potential to improve laboratory efficiency, reduce subjectivity, and standardize complex analytical processes. In the field of flow cytometry, particularly for the detection of minimal/measurable residual disease (MRD), deep neural networks (DNNs) offer a promising approach to enhance the reliability and consistency of data analysis while reducing the considerable time burden on laboratory staff [1].

MRD status in patients with chronic lymphocytic leukemia (CLL) has become increasingly important as it is not only a powerful independent predictor of survival but also plays a potential role in MRD-directed therapies [1,2,3,4,5]. Flow cytometric (FC) immunophenotyping is critical in detecting MRD in CLL patients, but traditional flow cytometric analysis is complex, time-consuming, and prone to subjectivity [6,7,8,9,10,11]. The analysis of multi-parameter flow cytometry data for MRD detection represents a significant bottleneck in the laboratory workflow, typically requiring 10–15 min per case for manual analysis by experienced technologists [8]. The integration of AI-assisted analysis offers the potential to dramatically reduce this time while maintaining or potentially improving analytical performance.

Clinical implementation of AI-assisted flow cytometry analysis requires a two-phase approach: rigorous clinical validation before deployment followed by continuous performance monitoring after implementation. For clinical validation, comprehensive assessment of method comparison, precision, and analytical sensitivity is essential to ensure the AI-assisted approach meets or exceeds the performance of current manual analysis by a trained human. This includes quantitative comparison against expert manual analysis (manual reference method), verification of precision across multiple operators and instruments, and confirmation of analytical sensitivity at clinically relevant MRD levels as specified in the Clinical Laboratory Improvement Amendments (CLIA) [12]. Only through such thorough validation can we establish confidence in the AI-assisted approach for clinical use.

However, successful validation represents only the beginning of quality assurance for AI/ML systems in clinical laboratories. As highlighted in recent publications addressing AI implementation in healthcare, without consistent and rigorous performance monitoring, the benefits of AI/ML could be overshadowed by inaccuracies, inefficiencies, and potential patient harm [13,14]. AI models, being inherently data-driven, are susceptible to performance degradation over time due to factors, such as covariate shift (changes in input data distribution) or concept drift (changes in the relationship between input data and target output) [15,16].

For an AI-assisted flow cytometry approach to CLL MRD detection, potential sources of covariate shift include variations in instrument performance, reagent lots, sample preparation procedures, and patient-specific differences. Such shifts can lead to deteriorating model performance if not detected and addressed promptly. Therefore, comprehensive performance monitoring strategies—including daily electronic quality control, input data drift detection, real-time error analysis, and statistical acceptance sampling—are essential for maintaining confidence in the AI-assisted approach over time and ensuring compliance with regulatory requirements, such as those specified by CLIA [12,17].

Our group previously developed a DNN-assisted approach for automated CLL MRD detection using a 10-color flow cytometry panel from 202 consecutive CLL patients who had completed therapy, collected between February 2020 and May 2021 [8]. This development cohort comprised 142 CLL MRD-positive and 60 CLL MRD-negative samples, with more than 256 million total events analyzed in an 80:10:10% ratio for training, validation, and testing, respectively. The DNN was trained on uncompensated data from a single-tube panel consisting of CD5-BV480, CD19-PE-Cy7, CD20-APC-H7, CD22-APC, CD38-APC-R700, CD43-BV605, CD45-PerCP-Cy5.5, CD200-BV421, Kappa-FITC, and Lambda-PE. The 10-color panel has been extensively validated for its robustness and reliability of detecting CLL MRD. Our initial work demonstrated high accuracy in detecting CLL MRD and potential for significant workflow improvements, based on an average DNN inference time of 12 s to process 1 million events per case through the model. In the current study, we describe the comprehensive clinical laboratory validation and post-implementation performance monitoring of the DNN-assisted human-in-the-loop workflow for CLL MRD detection by flow cytometry (Figure 1), ensuring that the benefits of AI integration are realized without compromising patient care.

## 2. Methods

### 2.1. Method Comparison

A method comparison study was conducted to evaluate the performance of DNN-assisted human-in-the-loop analysis for CLL MRD detection (DNN + 2nd review) compared to the manual reference method. The sample size was calculated based on both binary classification parameters (minimum tolerable accuracy of 97.5%, alpha of 5% representing type I error rate, beta of 20% representing type II error rate or a statistical power of 80%) and quantitative method comparison via Deming regression (expected correlation coefficient of 0.95, tolerable slope deviation of ±0.05), yielding a minimum requirement of 240 consecutive clinical samples based on the quantitative approach. The sample set included both MRD-positive and MRD-negative specimens, with at least 30% in each category, and with MRD-positive samples spanning the analytical range.

Ground truth for each case was established at both the qualitative level (positive vs. negative) and quantitative level (percentage of clonal events) by an on-bench technologist and secondary review by a member of the development team using the current validated manual reference approach in Infinicyt software (v 2.1) [Cytognos, Spain]. For the DNN approach, a hybrid workflow was implemented using both a full-cohort DNN (F-DNN) for initial analysis of all cases and a low-count DNN (L-DNN) specifically for cases with predicted low event counts (<1000 events) as described by Salama et al. [8].

The acceptance criteria for the primary analysis were as follows: (1) observed accuracy for binary classification not significantly lower than the minimum tolerable accuracy of 97.5% (alpha = 5%) with negative predictive value (NPV) above 97.5%; and (2) observed regression slope deviation not significantly higher than 0.05.

### 2.2. Precision

#### 2.2.1. Quantitative Precision

To verify the precision of the DNN + 2nd review method, raw flow cytometry standard (FCS) files from the original method evaluation study were reanalyzed. Six samples representing clinically relevant CLL MRD levels (5 CLL MRD-positive bone marrow and 1 CLL MRD-positive peripheral blood sample, all with <1% clonal events) were included in the analysis. No new sample processing or acquisition was performed, as this precision verification study aimed to verify that the human-in-the-loop AI model maintains the precision claims established in the original method evaluation.

Each sample was assayed in triplicate on two different instruments, and analyses were performed by two operators using the DNN + 2nd review method, generating a total of 12 replicates per sample (3 replicates × 2 instruments × 2 operators). Each operator analyzed 36 FCS files (3 replicates × 2 instruments × 6 samples).

The acceptance criteria required verification of precision estimates (repeatability and within-laboratory SD) from the original method evaluation study, with specific upper verification limits established for each sample as shown in the Appendix A.

#### 2.2.2. Qualitative Precision

To verify the qualitative precision of the DNN + 2nd review method across multiple operators, we designed a blinded study using a set of historical cases. Ten CLL MRD-positive samples representing a mixture of clonal percentages and four CLL MRD-negative samples were selected for analysis. Each file was replicated four times, and the files were de-identified and renamed to aid with blinding. This created a set of 56 blinded FCS files (14 samples × 4 replicates) that were randomly sorted and then analyzed by four trained technologists over a period of two weeks to reduce the possibility of recall for the four replicates per case. The qualitative precision data files were interspersed with the method comparison study to further aid with blinding.

The total number of analyses performed was 224 (14 samples × 4 replicates × 4 technologists). For each sample, we assessed the level of agreement among the four technologists by calculating accordance, concordance, and concordance odds ratio as described by Langton et al. [18]. Additionally, a test of equal sensitivities and specificities across all four raters was performed following the methodology described by Wilrich [19].

The acceptance criterion for this study was that there should be no significant difference in sensitivity and specificity of identifying CLL MRD levels across the four technologists when using the DNN + 2nd review method (alpha = 0.05).

### 2.3. Analytical Sensitivity (Method Detection Limit)

The lower level of quantitation (LLOQ) of the assay is 0.002% (2 × 10^−5^) based on 1,000,000 total events analyzed and an abnormal cell immunophenotype detected in a cluster of at least 20 cells. The limit of detection (LOD) is 0.001% (1 × 10^−5^). The assay sensitivity meets or exceeds the 0.01–0.001% (10^−4^–10^−5^) level of detection by flow cytometry, as recommended by the NCCN and iwCLL guidelines for MRD analysis in CLL [20,21]. Results between the LOD and LLOQ may be reported as suspicious or equivocal.

To verify the limit of detection and quantification, the raw FCS files from the original LLOQ evaluation study were reanalyzed. Two CLL MRD-positive bone marrow specimens and one CLL MRD-positive peripheral blood specimen, previously diluted to targeted clonal percentages of 0.02% (1 replicate each), 0.002% (3 replicates each), and 0.001% (3 replicates each), were included in this assessment. No new sample processing or acquisition was performed, as this verification study aimed to confirm that the human-in-the-loop AI model achieves the precision claims from the original method evaluation study at the LLOQ.

One technologist evaluated each FCS file from the original LLOQ study using the DNN + 2nd review method. The acceptance criteria specified that the calculated intra-assay CV for each targeted 0.002% MRD triplicate should be <30% to verify the LLOQ claims.

### 2.4. Real-Time Performance Monitoring

Following the implementation of the DNN-assisted analysis pipeline for CLL MRD detection, we established a comprehensive monitoring system to ensure continuous quality assessment. This system operates in four main ways: daily electronic quality control, input data drift detection, error analysis monitoring, and attribute acceptance sampling for negative cases.

#### 2.4.1. Daily Electronic Quality Control

At the beginning of each user’s shift, a prepared test file undergoes complete processing through both stage 1 (generate event-level DNN-inferences) and stage 2 (human review and saving of corrected event-level classifications). The system verifies that the workflow is functioning correctly by comparing the hashed file contents against reference values established during validation. This provides immediate confirmation that the DNN script is being executed as expected for that specific user’s workstation and the local Infinicyt software configuration. Any deviations from expected hash values trigger alerts, allowing for immediate troubleshooting before clinical sample analysis begins.

#### 2.4.2. Input Data Drift and Out-of-Distribution Detection

To identify potential changes in the characteristics of input data that could impact DNN performance, we implemented a drift detection system based on a data reconstruction approach using Principal Component Analysis (PCA) to identify subtle distributional changes not apparent at the individual feature level. The three-step process begins with data preparation (missing value imputation, frequency encoding, and standardization), followed by dimensionality reduction where PCA captures approximately 65% of variance from the training cohort, creating a compressed latent space representation. Finally, data are reconstructed using inverse PCA transformation, and Euclidean distances between original and reconstructed data points are calculated to produce a reconstruction error metric. Significant deviations in this error (exceeding three standard deviations from the reference mean) indicate potential drift, allowing detection of complex, multivariate shifts that might adversely affect model performance even when individual features appear stable. The metric can also be used for out-of-distribution detection to identify new samples that are different from the training distribution.

For each analyzed specimen, a random subset of 10,000 events was extracted and compared against a reference data distribution derived from the training dataset. The system calculates upper and lower threshold boundaries based on the training set distribution (mean ± 3 standard deviations). Each analyzed sample receives a quantitative drift score and an alert status (True/False) indicating whether the sample falls outside the expected range. These values are logged to a database and visualized on a monitoring dashboard, allowing laboratory staff to identify trends or sudden shifts in data characteristics that might require investigation or model recalibration. Input data drift is formally defined as an occurrence when 5 or more samples processed on the same calendar day exhibit reconstruction error values that consistently deviate from the expected reference range in the same direction. Specifically, all deviating samples must fall either exclusively above the upper threshold or exclusively below the lower threshold to be considered a true drift event, rather than random variation.

#### 2.4.3. Error Analysis Monitoring

For ongoing performance verification, the laboratory monitors the concordance between the classification of individual events from the DNN output (prior to human review) against the final corrected classifications (DNN + 2nd review). Each event is categorized as:•True Positive (TP): Correctly classified as clonal event by DNN and verified by technologist•True Negative (TN): Correctly classified as non-clonal event by DNN and verified by technologist•False Positive (FP): Incorrectly classified as clonal event by DNN but corrected by technologist•False Negative (FN): Incorrectly classified as non-clonal event by DNN but corrected by technologist

The system calculates key performance metrics: accuracy, sensitivity, specificity, positive predictive value (PPV) and negative predictive value (NPV) for the clonal event population versus other events. These metrics are logged to a relational database and displayed graphically on a real-time dashboard. While no specific cutoff values are established for corrective action at this time, this monitoring system enables identification of performance trends and potential degradation over time, particularly with changes in reagent lots, instrument calibration, or sample preparation protocols.

#### 2.4.4. Attribute Acceptance Sampling for Negative Cases

Ready identification of false-positive AI classifications is possible in a human-in-the-loop workflow, since human reviewers can pay special attention to events classified as clonal. However, identification of false-negative AI classifications is more challenging, since the clonal events could be classified as any of the remaining normal classes. To provide additional quality assurance focused specifically on the critical performance characteristic of false negative results, we implemented a periodic review process based on attribute acceptance sampling principles. This approach focuses on confirming the reliability of negative case determinations from the DNN + 2nd review workflow through targeted expert review.

Based on our laboratory’s average volume of approximately 74 negative cases per month (determined by DNN + 2nd review), we established a semi-annual quality control review cycle covering approximately 444 negative cases per 6-month period. Using the hypergeometric distribution, we calculated that to achieve 95% confidence that at least 95% of reported “negative” cases are truly negative, a random sample of 84 “negative” cases must be drawn from each 6-month period, with an acceptance criterion of no more than 1 allowable failure in the sample.

To distribute this workload evenly, 14 randomly selected negative cases undergo additional expert review each month. The results of these reviews are documented and analyzed at the end of each 6-month cycle. If more than one false negative is identified within the 84-case sample, a comprehensive investigation is initiated, potentially leading to workflow adjustments or consideration of model re-verification or retraining.

### 2.5. Analysis Time Reduction Study

To evaluate the long-term impact of the DNN + 2nd review workflow on laboratory efficiency, we conducted a timing study across multiple laboratory technologists during routine clinical operations. This study was designed to quantify real-world time savings in comparison to our pre-implementation baseline measurements and to assess the consistency of these efficiency gains over time.

Five laboratory technologists with varying levels of experience participated in the timing study over a one-month period. For each case analyzed using the DNN + 2nd review method, technologists recorded the total analysis time from initiation of review to completion of the final report. The start time was defined as the moment when the technologist began reviewing the DNN-classified events, and the end time was recorded when the technologist finalized their review and classification.

No specific case selection criteria were applied; all consecutive clinical samples received during the study period were included, representing the typical case mix encountered in routine practice. The recorded times were compared against two reference points: (1) pre-implementation baseline measurements of manual analysis time collected before the DNN-assisted workflow was deployed, and (2) the initial post-implementation measurements. For statistical analysis, we calculated the mean, standard deviation, minimum, and maximum analysis times for the overall dataset and for each individual technologist. The percentage reduction in analysis time compared to pre-implementation baseline was calculated to quantify the workflow efficiency gain.

## 3. Results

### 3.1. Method Comparison

#### 3.1.1. Binary Classification Accuracy

As shown in Appendix A, the DNN + 2nd review approach demonstrated high concordance with the manual reference method, correctly classifying 234 of 240 samples (97.5% overall accuracy). The approach showed 100% sensitivity (129/129) and 94.6% specificity (105/111). The positive predictive value was 95.6% (129/135) and the negative predictive value was 100% (105/105). For the 51 bone marrow samples analyzed, 49 (96.1%) were correctly classified, and for the 189 peripheral blood samples analyzed, 185 (97.9%) were correctly classified.

#### 3.1.2. Quantitative Method Comparison

The quantitative comparison demonstrated excellent correlation between the DNN + 2nd review and manual reference methods (Figure 2A, Appendix A). The correlation coefficient from Deming regression was 0.99, exceeding the acceptance criterion of 0.95. The regression slope was 0.99, within the tolerable deviation range of ±0.05 (0.95–1.05). The study population included an appropriate distribution of both MRD-positive cases (129/240, 53.8%) and MRD-negative cases (111/240, 46.2%), with both categories exceeding the minimum requirement of 30%. The Deming regression slopes and intercepts for the bone marrow and peripheral blood samples did not differ significantly (slope of 0.996 vs. 0.991, *p* = 0.49 and intercept of −0.016 vs. 0.004, *p* = 0.37).

As a descriptive analysis, we also evaluated the performance of the DNN model alone (without human review, DNN-only) compared to the manual reference method (Figure 2B). This analysis showed a good but slightly lower correlation coefficient of 0.97 and a Deming regression slope of 0.95. The overall accuracy was 90.4% with a negative predictive value of 94.0%. In one notable outlier case (highlighted in Figure 2B), expert analysis identified 418,567 abnormal events with an immunophenotype consistent with mantle cell lymphoma (MCL), which was confirmed by ancillary testing showing a CCND1/IGH fusion in approximately 77% of nuclei. The DNN, which was specifically trained on CLL immunophenotypes, misclassified 400,080 of these events as normal B-cells.

### 3.2. Precision

#### 3.2.1. Quantitative Precision

Six samples (five bone marrow and one peripheral blood) were analyzed in threereplicates on two instruments by two operators. As shown in Table 1, five of the six samples met all precision criteria, with measured standard deviations below the upper verification limits established in the verification plan. One sample (BM1) showed a repeatability standard deviation (0.06798) slightly above the upper verification limit (0.06636), but its within-laboratory standard deviation (0.07097) remained below the corresponding upper verification limit (0.16432).

Further investigation of sample BM1 revealed that the slightly elevated imprecision was attributable to differences in how operators interpreted the boundary between debris/aggregates and viable events. Visual examination of the flow cytometry plots showed that the DNN classified some events as debris and doublets, while identifying abnormal events. The two operators differed in their interpretation of these boundary events, with one operator including more of the borderline events as viable cells than the other. Importantly, despite the minor technical deviation for sample BM1, the difference in measured clonal percentages did not affect the ultimate clinical interpretation. Both operators correctly identified the sample as CLL MRD-positive, with a CD5+ B-cell population showing an immunophenotype consistent with CLL. Furthermore, the within-laboratory standard deviation for this sample remained within acceptable limits, indicating that the overall precision of the DNN + 2nd review method was acceptable for clinical use.

#### 3.2.2. Qualitative Precision

To evaluate the consistency of CLL MRD classification across multiple operators, we analyzed the results from the qualitative precision study involving 14 historical cases (10 MRD-positive and four MRD-negative) analyzed by four technologists in quadruplicate. There was no significant difference in diagnostic performance across the four technologists when using the DNN + 2nd review method. The test of equal sensitivities yielded a *p*-value of 0.389, the test of equal specificities showed a *p*-value of 1.000, and the test of equal accuracies resulted in a *p*-value of 0.390. All *p*-values were well above the significance threshold of 0.05 for identifying a significant difference.

For MRD-positive samples, the average accordance (intra-observer agreement) was 98.75%, and the average concordance (inter-observer agreement) was also 98.75%, with a concordance odds ratio of 1. For MRD-negative samples, both accordance and concordance were 100%, with a concordance odds ratio of 1.

### 3.3. Analytical Sensitivity (Method Detection Limit)

Three samples (two CLL MRD-positive bone marrow specimens and one CLL MRD-positive peripheral blood specimen) were diluted to targeted clonal percentages and analyzed using the DNN + 2nd review method. The DNN + 2nd review method successfully identified clonal populations at the 0.002% level in all three samples (PB1, BM1, and BM2). The calculated intra-assay coefficient of variation (CV) for the 0.002% triplicates ranged from 6.57% to 23.52% across the three samples, all well below the acceptance criterion of <30%.

### 3.4. Real-Time Performance Monitoring

#### Input Data Drift Detection

During the first 6-month monitoring period from May 2024 to November 2024, a total of 874 clinical samples were analyzed, with 26 samples (2.97%) triggering drift alerts. The formal drift criterion of five or more samples from the same day consistently deviating from the expected range in the same direction was not met during this 6-month period, indicating a stable process. The observed alerts were sporadic and primarily associated with individual outlier cases rather than systematic shifts in the input data distribution. These alerts all occurred when the sample’s reconstruction error value exceeded the established upper threshold of 1.7589. Further investigation of the 26 alert cases revealed several distinct patterns explaining the deviation from the expected input distribution (Table 2). The majority of alerts (17/26, 65.4%) were associated with high-burden CLL cases (>40% abnormal cells), which were underrepresented in the training dataset that was enriched for low-level MRD cases. These high-burden cases demonstrated reconstruction error values ranging from 1.85 to 3.99, with higher values generally correlating with higher tumor burden.

The remaining alerts were triggered by samples containing non-CLL clonal populations with distinct immunophenotypes from those in the training data. These included mantle cell lymphoma (MCL) cases (3/26, 11.5%), CD5-negative B-cell lymphoproliferative disorder cases (4/26, 15.4%), and cases with unusual immunophenotypic features (2/26, 7.7%). Figure 3 displays the reconstruction error values for each clinical sample (dots) analyzed through the DNN-assisted workflow during a 1-month period.

### 3.5. Error Analysis Monitoring

During the first 6-month monitoring period, we analyzed a total of 874 clinical cases with the DNN + 2nd review approach. The DNN model alone (DNN-only) achieved a sensitivity of 97% and a specificity of 59% compared to the final classifications after technologist review (DNN + 2nd review). The positive predictive value was 73%, while the negative predictive value was 95%. As shown in Figure 4, there was strong correlation between the DNN model and the DNN + 2nd review approach (r = 0.977), with a Deming regression slope of 0.98.

The analysis revealed that the DNN model missed 13 positive cases (1.5% of total cases) that were subsequently identified during technologist review. Detailed examination of these cases revealed distinct immunophenotypic patterns that are summarized in Table 3 and Appendix A. Of the 13 missed cases, five were CD5-negative B-cell lymphoproliferative disorders, which differ significantly from the classic CLL immunophenotype used to train the DNN model. The remaining eight cases were CD5-positive, but seven of these displayed immunophenotypic features atypical for CLL, including moderate (rather than dim) CD20 and CD22 expression, negative or partial CD43 expression, normal CD45 expression, moderate light chain expression, and in four cases, negative CD200 expression. Only one case with a classic CLL immunophenotype was “missed” by the DNN-only analysis. In this case, the DNN model detected 17 true CLL events, which was below the 20-event cutoff for positivity. Representative immunophenotypic features of missed cases by the DNN model are shown in Figure 5.

### 3.6. Attribute Acceptance Sampling for Negative Cases

During the first 6-month monitoring period (from May 2024 to November 2024), we selected and re-analyzed 84 cases that had originally been reported as negative by the DNN + 2nd review workflow. The re-analysis was performed by three senior laboratory staff who had not participated in the original review process. Of the 84 cases re-analyzed, 83 (98.8%) were confirmed as true negatives, with only one case (1.2%) identified as a false negative, which carried CD5-negative immunophenotype. This result met our acceptance criterion of no more than one allowable failure in the sample set. This high level of agreement (98.8%) provides statistical confidence at the 95% level that at least 95% of all reported negative cases are true negatives.

### 3.7. Analysis Time Reduction

The timing study captured data for 161 cases analyzed across five technologists using the DNN + 2nd review method during routine clinical operations. As shown in Table 4, the average analysis time was 4.2 ± 2.3 min (range: 1.0–10.5 min). This represents a 60.3% reduction compared to the pre-implementation manual analysis time of 10.5 ± 5.8 min (range: 3.5–27.5 min) established in our baseline workflow assessment.

## 4. Discussion

The results of our comprehensive validation and post-implementation monitoring demonstrate that the DNN-assisted human-in-the-loop approach for CLL MRD detection provides accurate, precise, and reliable results while dramatically reducing analysis time compared to traditional manual methods. Our findings validate the approach described by Salama et al. [8] and extend our laboratory’s work by establishing a robust framework for ongoing performance monitoring that ensures the continued safety and effectiveness of this AI-assisted workflow in routine clinical practice.

The method comparison study demonstrated excellent concordance between the DNN + 2nd review approach and the manual reference method relying on expert analysis, with 97.5% overall accuracy for qualitative determination and a correlation coefficient of 0.99 for quantitative assessment. The quantitative comparison showed no significant systematic bias, with a Deming regression slope of 0.99, well within our predefined acceptance criteria. These findings indicate that the DNN-assisted approach provides essentially equivalent diagnostic information to the current gold standard method while significantly improving laboratory workflow efficiency.

This efficiency improvement is clearly demonstrated by the timing study, which showed a consistent reduction in analysis time of approximately 60% compared to manual analysis (4.2 ± 2.3 min versus 10.5 ± 5.8 min), even after accounting for the additional quality assurance steps implemented to monitor for performance drift or shift over time. The persistence of this time reduction over a one-year period confirms that the efficiency gains are sustainable in routine clinical practice, not merely a short-term benefit during initial implementation. With an average time savings of 6.33 min per case and our laboratory’s annual volume of approximately 1500 CLL MRD cases, this translates to approximately 158 h of technologist time saved per year. This substantial efficiency improvement allows for better resource allocation, potentially reducing turnaround times and increasing laboratory capacity without requiring additional staffing. Similar efficiency improvements have been reported by other groups implementing AI-assisted workflows in clinical laboratory settings [22], suggesting that this benefit is a consistent feature of successful AI integration rather than an isolated finding.

Our precision studies confirmed that the DNN + 2nd review method maintains the high level of precision established in the original validation, with excellent repeatability and within-laboratory precision for both qualitative and quantitative assessments. The qualitative precision study demonstrated no significant differences in sensitivity or specificity across multiple technologists, with near-perfect accordance and concordance statistics. The quantitative precision study showed that standard deviations for repeatability and within-laboratory precision were below the established upper verification limits for all but one sample, confirming that the DNN-assisted method achieves precision comparable to the original manual method. These findings align with previous studies demonstrating that AI-assisted approaches can reduce inter-operator variability in complex diagnostic tasks [23,24].

The analytical sensitivity verification confirmed the method’s ability to reliably detect and quantify CLL MRD at levels as low as 0.002%, consistent with the clinical requirements for MRD monitoring in CLL patients following therapy. This level of sensitivity is essential for early detection of disease recurrence and for evaluating treatment efficacy, particularly with the advent of novel targeted therapies that can achieve deep molecular responses [25]. The importance of high analytical sensitivity in MRD detection has been emphasized by consensus guidelines and is critical for optimal patient management [20].

Beyond the initial validation, our implementation of a multi-faceted performance monitoring system has provided valuable insights into the real-world performance of the AI-assisted workflow and established a model for continuous quality assurance of AI applications in clinical laboratories. The input data drift detection system identified a small percentage of cases (2.97%) that fall outside the expected distribution, primarily high-burden CLL cases and non-CLL neoplasms with distinct immunophenotypes. This finding highlights the importance of understanding the limitations of AI models trained on specific disease entities and the value of automated mechanisms for identifying cases that may challenge the model’s classification capabilities, i.e., out-of-distribution examples [15,26,27].

The error analysis monitoring revealed specific patterns in the performance of the DNN-only, with a high sensitivity (97%) but more modest specificity (59%) when compared DNN + 2nd review, which is expected since the DNN was optimized for high negative predictive value. It is important to clarify that this specificity concern primarily relates to the DNN-only analysis rather than the DNN + 2nd review approach, which demonstrated excellent performance in clinical validation (97.5% accuracy, 100% sensitivity, 94.6% specificity). Our approach intentionally prioritizes high sensitivity and NPV, as false-positive classifications can be readily corrected during expert review, whereas false-negatives are extremely difficult to detect among millions of events. To improve specificity while maintaining sensitivity, we are exploring several strategies, including gradient boosting methods, uncertainty quantification techniques, class weightings, and focal loss functions. It is also possible that the incorporation of other markers, such as CD79b or T cell markers, could improve the performance of the model. The analysis of missed cases identified specific immunophenotypic features that challenge the DNN’s classification abilities, particularly CD5-negative B-cell neoplasms (5/13) and lymphoproliferative disorders with atypical immunophenotypes (7/13), with only one classic CLL case missed due to low event counts just below our threshold. This information is valuable for both targeted education of laboratory staff and potential future refinements of the DNN model to improve its performance on these challenging edge cases. Similar error pattern analyses have been proposed to maintain patient safety at the forefront of AI development and facilitate safe clinical deployment [28].

The attribute acceptance sampling program provided statistical verification that at least 95% of reported negative cases are true negatives (with 95% confidence), further validating the reliability of the DNN + 2nd review workflow for making negative determinations in routine clinical practice. This approach offers an efficient mechanism for ongoing verification of this critical performance characteristic without requiring exhaustive manual review of all negative cases, adapting established quality control techniques from manufacturing to the clinical laboratory setting [29].

Together, these monitoring components form a comprehensive quality assurance framework that addresses the unique challenges of maintaining AI performance in clinical settings. AI models are inherently data-driven and susceptible to performance degradation over time due to factors, such as covariate shift or concept drift. Our monitoring framework directly addresses these challenges by incorporating mechanisms for detecting input data drift, tracking error patterns, and statistically verifying critical performance characteristics over time.

Bazinet et al. recently published their experience using DeepFlow, a commercially available AI-assisted FC analysis software for CLL MRD detection. They reported overall concordance of 96% between their AI-assisted analysis and expert manual gating, but a lower correlation for quantitative results of 0.865 compared to our workflow [30]. Interestingly, they encountered similar challenges in cases with atypical immunophenotypes, particularly CLL with trisomy 12, which showed higher expression of B cell markers and lower CD43 expression that made AI classification more difficult. While the DeepFlow software offers a user-friendly graphical interface and rapid processing speed (3 s for ~1 million events), our DNN-assisted approach offers several advantages. Our model was specifically designed for implementation within existing clinical laboratory workflows, with complete integration into the Infinicyt software platform widely used in flow cytometry laboratories. Additionally, our comprehensive post-implementation monitoring system—including daily electronic quality control, input data drift detection, error analysis monitoring, and attribute acceptance sampling—provides ongoing safeguards that are essential for maintaining reliable performance in routine clinical use. Both studies highlight the potential of AI to dramatically improve laboratory efficiency while maintaining diagnostic accuracy, with each approach offering different advantages for implementation based on a laboratory’s specific needs and existing infrastructure. The human-in-the-loop design of both workflows represents a balanced approach to AI integration in clinical laboratories, leveraging the efficiency and standardization benefits of automation while maintaining expert oversight for cases that may challenge the AI model. This aligns with emerging best practices for clinical AI applications, which recognize that human expertise and AI capabilities are often complementary rather than competitive.

The limitations of our study include its focus on a specific disease entity (CLL) and flow cytometry panel at a single institution, which limits the generalizability of our findings to other applications and laboratories. Due to the inherent pre-analytical and analytical variability associated with instrument–reagent combinations and panel design, we expect that each laboratory would need to train and validate their own AI model with local data to account for site-specific variables, including instruments, reagents, and pre-analytic factors. Additionally, while our post-implementation monitoring has extended over several months, longer-term follow-up will be needed to fully assess the stability of the DNN model’s performance over time and across changing clinical and laboratory conditions. Whether model retraining can address performance degradation over time is also an unresolved question. Our DNN model was trained on post-treatment samples, enabling effective recognition of common treatment-induced immunophenotypic changes, such as CD20 negativity and dim CD23 expression. Error analysis revealed that missed cases were primarily attributable to atypical baseline immunophenotypes, rather than treatment-related changes. While our current model handles typical post-treatment modifications well, emerging therapies, such as CAR T-cell therapy, could introduce new challenges for flow cytometric CLL MRD detection.

Future work should focus on the development of more sophisticated drift detection algorithms that can differentiate between different types of drift (e.g., covariate shift versus concept drift), which could further refine our monitoring approach and provide more targeted guidance for model maintenance and updating [31]. A promising avenue also involves the integration of uncertainty quantification methods to improve the detection and handling of atypical immunophenotypes. By implementing techniques, such as Monte Carlo dropout, deep ensembles, or Bayesian neural networks, we could quantify the model’s epistemic uncertainty (uncertainty due to limited knowledge) for each analyzed event and case [32,33,34,35]. High uncertainty values would automatically flag cases with unusual or ambiguous immunophenotypic features for closer expert review, potentially improving the detection of CD5-negative lymphoproliferative disorders and other non-CLL neoplasms that currently challenge the DNN. This technique would further enhance the human-in-the-loop workflow by focusing expert attention on the most challenging cases while allowing the DNN to operate independently on typical cases with high confidence, potentially driving additional efficiency improvements while maintaining or enhancing diagnostic accuracy.

## 5. Conclusions

Our results demonstrate that a DNN-assisted human-in-the-loop workflow for CLL MRD detection by flow cytometry can achieve diagnostic performance equivalent to expert manual analysis while dramatically reducing analysis time. The implementation of a comprehensive performance monitoring system ensures the ongoing safety and effectiveness of this approach in routine clinical practice. This work provides a model for the responsible integration of AI technologies in clinical laboratories, balancing the efficiency benefits of automation with the critical oversight provided by human expertise. As AI applications continue to expand in laboratory medicine, such balanced approaches and rigorous performance monitoring frameworks will be essential for ensuring that these technologies enhance rather than compromise the quality and safety of patient care.

## Figures and Tables

**Figure 1 cancers-17-01688-f001:**
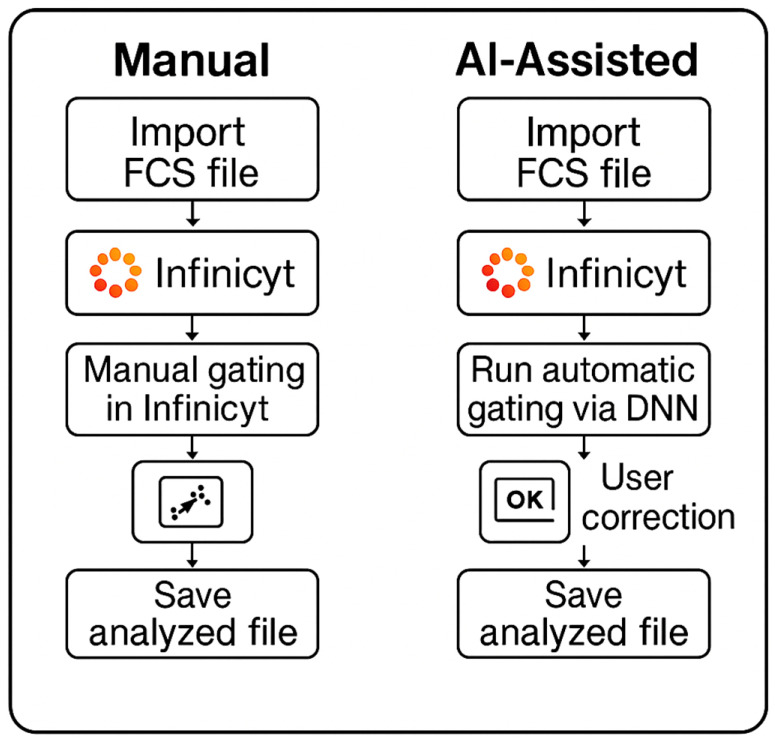
**Comparison of manual and AI-assisted workflows for flow cytometry analysis in CLL MRD.** This flowchart illustrates the key steps in two approaches used to analyze flow cytometry standard (FCS) files for chronic lymphocytic leukemia (CLL) minimal/measurable residual disease (MRD) assessment within the Infinicyt software (v2.1) environment. In the manual workflow, FCS files are imported into Infinicyt, gated manually by the user, and saved after interpretation. In the AI-assisted workflow, FCS files are also imported into Infinicyt, where a deep neural network (DNN) is launched via a scripting interface to perform automatic gating; the user then reviews and corrects the results as needed before saving the final analyzed file. Both workflows operate within Infinicyt but differ in the level of automation and required user input. The AI-assisted method enables more efficient case processing while maintaining human oversight.

**Figure 2 cancers-17-01688-f002:**
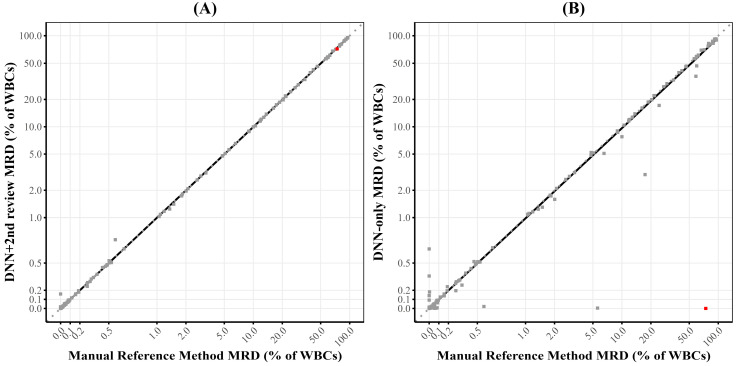
**Correlation between manual analysis (ground truth) and DNN-based approaches for quantitative MRD assessment.** (**A**) Scatter plot showing the correlation between the manual reference method and DNN + 2nd review method for the percentage of clonal events per analyzed white blood cells (WBCs), with a correlation coefficient of 0.99 and slope of 0.99. (**B**) Scatter plot showing the correlation between ground truth manual analysis and DNN-only analysis. The Deming regression shows a correlation coefficient of 0.970 and slope of 0.95. The red dot highlights an outlier case with mantle cell lymphoma that was misclassified by the DNN (**B**), and was subsequently corrected by the expert upon 2nd review (**A**).

**Figure 3 cancers-17-01688-f003:**
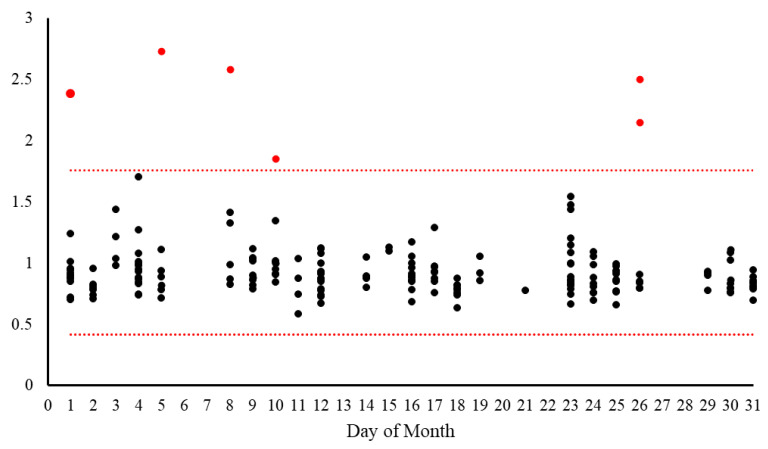
**Input data drift monitoring over time.** Reconstruction error values (*y*-axis) for each clinical sample (black and red dots) analyzed through the DNN-assisted workflow during a 1-month period, with the upper (1.7589) and lower (0.4147) thresholds shown as horizontal lines. Red dots indicate samples that triggered alerts by exceeding the upper threshold.

**Figure 4 cancers-17-01688-f004:**
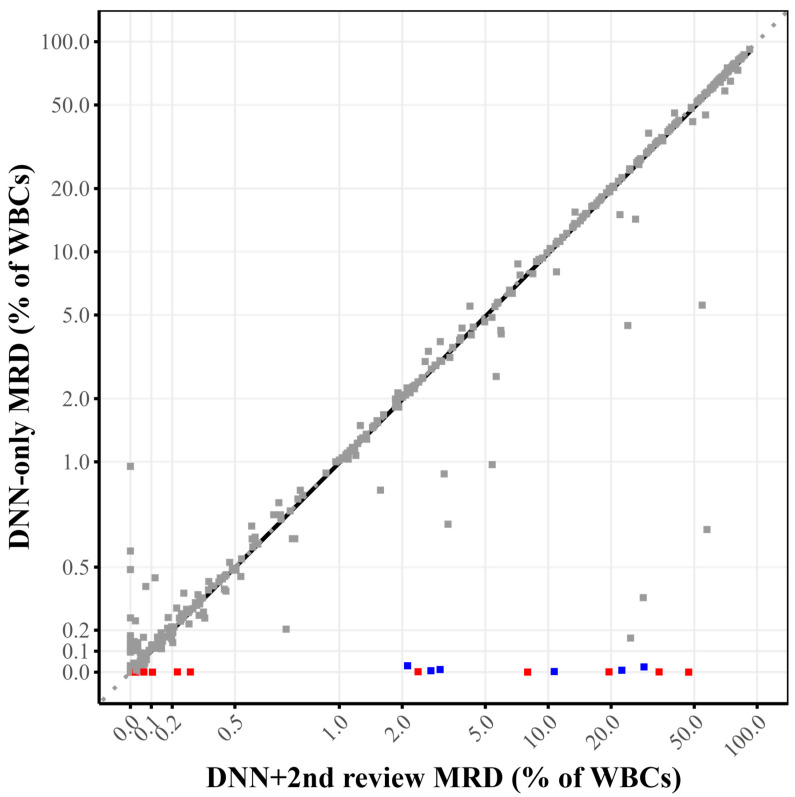
**Correlation between DNN-only and DNN + 2nd review approaches for quantitative MRD assessment.** Scatter plot showing the relationship between the percentage of abnormal events detected by DNN model alone (DNN-only, *y*-axis) versus DNN + 2nd review (*x*-axis) for the percentage of clonal events per analyzed white blood cells (WBCs), with a correlation coefficient of 0.98 and Deming regression slope of 0.98. Red dots on the *x*-axis represent the 13 missed cases where the DNN identified <20 clonal events, while blue dots represent cases where the DNN identified a clonal population ≥20 events, but there was a discrepancy between the DNN-only prediction (<0.1% MRD) and the DNN + 2nd review (>1%).

**Figure 5 cancers-17-01688-f005:**
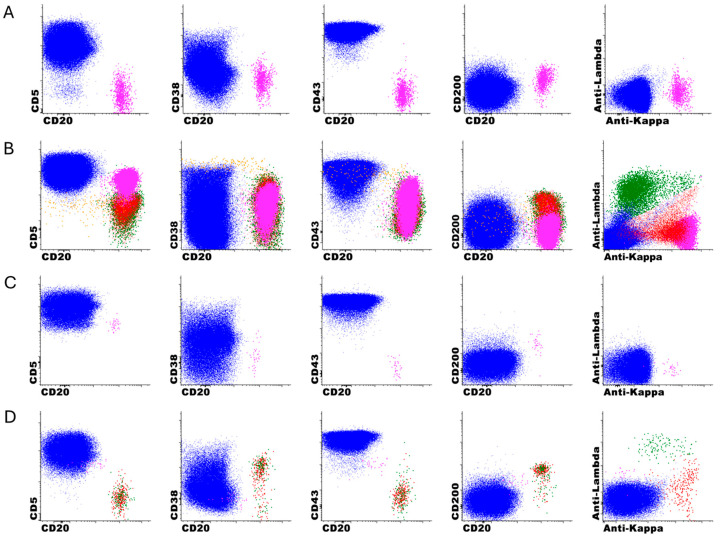
**Representative immunophenotypic features of missed cases by DNN-only analysis.** Clonal B cells in pink, polytypic kappa B cells in red, polytypic lambda B cells in green, T cells in blue, hematogones in orange. (**A**) CD5-negative clonal B cells; (**B**) CD5-positive clonal B cells, non-CLL-type; (**C**) CD5-positive clonal B cells, non-CLL-type; (**D**) CD5-positive clonal B cells, CLL type. The DNN-only detected 17 CLL events (less than 20-event cutoff for positivity, i.e. indetermindate or equivocal).

**Table 1 cancers-17-01688-t001:** Repeatability and within-laboratory precision for quantitative MRD assessment. SD: standard deviation; CD: coefficient of variation.

Sample	Mean (MRD %)	Repeatability SD	Repeatability CV	Within Laboratory SD	Within Laboratory CV	Upper Verification Limit—Repeatability SD	Upper Verification Limit—Within Laboratory SD
BM1	0.75262	0.06798	9.00%	0.07097	9.40%	0.06636	0.16432
BM2	0.37903	0.01444	3.80%	0.02269	6.00%	0.01919	0.03510
BM3	0.36078	0.01682	4.70%	0.01906	5.30%	0.02735	0.02528
BM4	0.55410	0.03104	5.60%	0.03104	5.60%	0.04622	0.06913
BM5	0.67446	0.05677	8.40%	0.05926	8.80%	0.06911	0.11944
PB1	0.02114	0.00337	16.00%	0.00399	18.90%	0.00667	0.00616

**Table 2 cancers-17-01688-t002:** Categorization of drift alert cases by underlying cause.

Alert Category	Count	Percentage	Drift Value Range
High-burden CLL (>40%)	17	65.4%	1.85–3.99
Mantle cell lymphoma	3	11.5%	1.78–2.51
CD5-negative B-cell disorders	4	15.4%	1.92–2.95
Atypical CLL immunophenotype	2	7.7%	1.79–1.84

**Table 3 cancers-17-01688-t003:** Summary of immunophenotypic features of 13 cases missed by DNN-only analysis.

	Number (% of 874 Cases)
Missed cases by DNN-only	13 (1.5%)
CD5-negative	5 (0.6%)
CD5-positive	8 (0.9%)
Non-CLL	7
CD20 moderate	7
CD22 moderate	7
CD43-negative or partial	7
CD45 not dim	7
Light chain moderate	7
CD200-negative	4
CLL	1 *

Note: DNN, deep neural network; CLL, chronic lymphocytic leukemia; *, not a complete miss as DNN model identified 17 CLL events (equivocal range).

**Table 4 cancers-17-01688-t004:** Analysis time comparison before and after implementation of the DNN-assisted workflow.

Analysis Method	Mean Time (min)	Standard Deviation (min)	Range (min)	Time Reduction
Pre-implementation manual analysis	10.5	5.8	3.5–27.5	—
Initial DNN-assisted workflow (go-live)	3.9	2.1	1.0–9.0	62.9%
Current DNN-assisted workflow (1-year follow-up)	4.2	2.3	1.0–10.5	60.3%

## Data Availability

Raw data were generated at the Mayo Clinic Cell Kinetics Laboratory. Derived data supporting the findings of this study are available from the corresponding author upon reasonable request.

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
