# Peer review of "Clinical Validation and Post-Implementation Performance Monitoring of a Neural Network-Assisted Approach for Detecting Chronic Lymphocytic Leukemia Minimal Residual Disease by Flow Cytometry"

_cancers, 2025, doi:10.3390/cancers17101688_

Round 1

Reviewer 1 Report

Comments and Suggestions for Authors

The paper by J.N. Seheult and Coworkers is a validation study on the clinical application of an Artificial Intelligence (AI)-based flow cytometric analysis of B-cell chronic lymphocytic leukemia measurable residual disease (B-CLL MRD).

This paper comes after a previous study from the same research group, in which the technical implementation and the feasibility of such an approach were described using a hybrid deep neural network model (Salama ME et al, ref.#8).

GENERAL COMMENTS

Deep neural network (DNN)-assisted approaches are a type of AI and machine learning being applied in the clinical laboratory, particularly in areas like cytology and flow cytometry on blood and bone marrow for diagnosing hematological malignancies. Models are specifically trained to differentiate between various hematological conditions based on flow cytometric data, such as distinguishing AML from ALL or MDS, and more recently also to detect and quantitate MRD.

The present paper shows that the implemented model of hybrid human-in-the-loop DNN is able to

provide diagnostic performance equivalent to expert manual analysis in B-CLL MRD detection, while substantially reducing analysis time. DNN alone seems however flawed by a modest 59% specificity, which detemines a large excess of false-positive cases, needing the correction by an expert supervisor.

The Authors should better clarify several points on this issue, namely if their DNN model is at present best suited to be of help in screening out negative cases, being endowed with a good negative predictive value, and how.

Moreover, the reason for this low specificity should be also discussed in greater detail, investigating the most repetitive phenotypic features that generate misinterpretation and false positive results, and why.

Clarifying this points can be of help to the readership, since DNN models are characterized by a 'black box' nature, preventing the users to fully understand the explanation of the decisional processes.

Once properly trained, a DNN-assisted approach becomes strictly panel- and disease immunophenotype-dependent, as correctly pointed out by the Authors. In the present study some experiments were accomplished using peripheral blood, while some other employed bone marrow cell suspensions. The two biological matrices are remarkably different, being characterized by a markedly diverse representation of the accompanying unrelevant cells, especially in case of a regenerating bone marrow. The Authors should address this point with an appropriate chapter, trying to elucidate if also a matrix-dependence can play a role in the assay performance and how to manage it, if present.

Immunophenotyping of B-CLL and B-CLL MRD is an issue still prone to revisions, and studies highlighting the advantages of alternative antibody panels are being frequently published (See: Rawstron AC, et al. Cytometry Part B (Clinical Cytometry) 2018; 94B: 121-128; and Salem DA, et al. Methods Mol Biol 2019; 2032: 311-321; and Chen X, et al. Cytometry Part B (Clinical Cytometry) 2024; 106: 181-191).

In the present study the Authors have used the 10-color antibody panel described in their previous paper (ref.#8) and reported in supplementary table 4. This 10-color antibody mixture does not include CD10, CD22 and T cell markers, as recommended -for instance - by the EuroFlow consortium.

The Authors should add a point in the discussion section on limitations of the study, addressing the possible improvement of the DNN-based system performance through the addition of some missing markers aimed at increasing specificity. Moreover, a few lines should be added, stressing that any change in the antibody panel - with time and scientific progress - may result in the need of an entire re-instruction and validation process of the DNN-based assay.

The Authors have established a lowest level of quantitation (LLOQ) of 0.002%, based on 20 B-CLL cells over 1 million clean CD45+ leukocytes, a sensitivity level typically used as the lowest level of detection (LLOD) instead (See: Illingworth A, et al.  Cytometry Part B (Clinical Cytometry) 2018; 94B: 49-66; and Sanoja-Flores L, et al. Blood Cancer Journal 2018; 8: 117). Please specify the respective established LLOD, if any, and what is the applied policy in the reporting if an analysis result gives a value higher than the LLOD but lower than the LLOQ.

In the previous paper on DNN in B-CLL MRD study (Ref.#8), the Authors have stated that "The hybrid DNN approach...reduced analysis time to approximately 12sec/case" versus 15-20 minutes/case for an experienced technologist. In the present paper the DNN-based "... analysis time was 4.17 ± 2.25 minutes (range: 1.0-10.5 minutes)", which is a remarkable variation. Is this variation taking into account the human-in-the-loop intervention to supervise flagged cases? This point should be adequately discussed.

SPECIFIC COMMENTS

Just a typo at line n. 52: ' predicotr of survivcal ' instead of 'predictor of survival'.

Methods: Please specify the meaning of the alpha and beta statistical parameters at line n. 101, for sake of clarity.

Author Response

***Comments1: Deep neural network (DNN)-assisted approaches are a type of AI and machine learning being applied in the clinical laboratory, particularly in areas like cytology and flow cytometry on blood and bone marrow for diagnosing hematological malignancies. Models are specifically trained to differentiate between various hematological conditions based on flow cytometric data, such as distinguishing AML from ALL or MDS, and more recently also to detect and quantitate MRD.

The present paper shows that the implemented model of hybrid human-in-the-loop DNN is able to provide diagnostic performance equivalent to expert manual analysis in B-CLL MRD detection, while substantially reducing analysis time. DNN alone seems however flawed by a modest 59% specificity, which detemines a large excess of false-positive cases, needing the correction by an expert supervisor. The Authors should better clarify several points on this issue, namely if their DNN model is at present best suited to be of help in screening out negative cases, being endowed with a good negative predictive value, and how. Moreover, the reason for this low specificity should be also discussed in greater detail, investigating the most repetitive phenotypic features that generate misinterpretation and false positive results, and why. Clarifying this points can be of help to the readership, since DNN models are characterized by a 'black box' nature, preventing the users to fully understand the explanation of the decisional processes.

***Response1: We have addressed these concerns in the Discussion on P15 L529-549:

“The error analysis monitoring revealed specific patterns in the performance of the DNN-only, with a high sensitivity (97%) but more modest specificity (59%) when compared to expert review, which is expected since the DNN was optimized for high negative predictive value. It is important to clarify that this specificity concern primarily relates to the DNN-only analysis rather than the DNN+2nd review approach, which demonstrated excellent performance in clinical validation (97.5% accuracy, 100% sensitivity, 94.6% specificity). Our approach intentionally prioritizes high sensitivity and NPV, as false-positive classifications can be readily corrected during expert review, whereas false-negatives are extremely difficult to detect among millions of events. To improve specificity while maintaining sensitivity, we are exploring several strategies including gradient boosting methods, uncertainty quantification techniques, class weightings, and focal loss functions. It is also possible that the incorporation of other markers such as CD79b or T cell markers could improve the performance of the model. The analysis of missed cases identified specific immunophenotypic features that challenge the DNN's classification abilities, particularly CD5-negative B-cell neoplasms (5/13) and lymphoproliferative disorders with atypical immunophenotypes (7/13), with only one classic CLL case missed due to low event counts just below our threshold. This information is valuable for both targeted education of laboratory staff and potential future refinements of the DNN model to improve its performance on these challenging edge cases. Similar error pattern analyses have been proposed to maintain patient safety at the forefront of AI development and facilitate safe clinical deployment [25].”

***Comments2: Once properly trained, a DNN-assisted approach becomes strictly panel- and disease immunophenotype-dependent, as correctly pointed out by the Authors. In the present study some experiments were accomplished using peripheral blood, while some other employed bone marrow cell suspensions. The two biological matrices are remarkably different, being characterized by a markedly diverse representation of the accompanying unrelevant cells, especially in case of a regenerating bone marrow. The Authors should address this point with an appropriate chapter, trying to elucidate if also a matrix-dependence can play a role in the assay performance and how to manage it, if present.

***Response2: We have now included summary statistics for the bone marrow and peripheral blood samples showing no difference in performance on P8 L305-307 and P8 L315-317:

“For the 51 bone marrow samples analyzed, 49 (96.1%) were correctly classified, and for the 189 peripheral blood samples analyzed, 185 (97.9%) were correctly classified.”

“The Deming regression slopes and intercepts for the bone marrow and peripheral blood samples did not differ significantly (slope of 0.996 vs 0.991, p = 0.49 and intercept of -0.016 vs 0.004, p = 0.37).”

***Comments3: Immunophenotyping of B-CLL and B-CLL MRD is an issue still prone to revisions, and studies highlighting the advantages of alternative antibody panels are being frequently published (See: Rawstron AC, et al. Cytometry Part B (Clinical Cytometry) 2018; 94B: 121-128; and Salem DA, et al. Methods Mol Biol 2019; 2032: 311-321; and Chen X, et al. Cytometry Part B (Clinical Cytometry) 2024; 106: 181-191). In the present study the Authors have used the 10-color antibody panel described in their previous paper (ref.#8) and reported in supplementary table 4. This 10-color antibody mixture does not include CD10, CD22 and T cell markers, as recommended -for instance - by the EuroFlow consortium. The Authors should add a point in the discussion section on limitations of the study, addressing the possible improvement of the DNN-based system performance through the addition of some missing markers aimed at increasing specificity. Moreover, a few lines should be added, stressing that any change in the antibody panel - with time and scientific progress - may result in the need of an entire re-instruction and validation process of the DNN-based assay.

***Response3: Our 10-color CLLMD panel is a modification of our original 8-color panel which had demonstrated clinical utility and had been utilized in our laboratory for years. We conducted extensive validation of the 10-color panel and confirmed that it offers greater sensitivity compared to the 8-color version. Additionally, we compared this 10-color panel with ClonoSEQ, an FDA-approved method for CLL MRD detection, and found a high concordance. During the panel design stage, we did evaluate the inclusion of CD79b, however it did not improve performance within our specific antibody combination. The panel already includes CD20. In summary, we are highly confident in the robustness and reliability of our 10-color CLL MRD panel. We have added this statement to the introduction section on P3 L95-97: “The 10-color panel has been extensively validated for its robustness and reliability of detecting CLL MRD.” We have also indicated in the Discussion on P15 L539-541 that performance could be improved by adding more markers: “It is also possible that the incorporation of other markers such as CD79b or T cell markers could improve the performance of the model.”

***Comments4: The Authors have established a lowest level of quantitation (LLOQ) of 0.002%, based on 20 B-CLL cells over 1 million clean CD45+ leukocytes, a sensitivity level typically used as the lowest level of detection (LLOD) instead (See: Illingworth A, et al.  Cytometry Part B (Clinical Cytometry) 2018; 94B: 49-66; and Sanoja-Flores L, et al. Blood Cancer Journal 2018; 8: 117). Please specify the respective established LLOD, if any, and what is the applied policy in the reporting if an analysis result gives a value higher than the LLOD but lower than the LLOQ.

***Response4: Lower limit of quantitation (LLoQ) is the ability to precisely and accurately measure low amounts of the measurand. The LLOQ of CLLMD is 0.002%. Lower limit of detection (LLoD) is the ability of the assay to distinguish signal from background. The LLoD = 0.001% for CLLMD. We have clarified the terminology and reporting strategy as follows on P5 L173-178: “The lower level of quantitation (LLOQ) of this assay is 0.002% (2 x 10-5) based on 1,000,000 total events analyzed and an abnormal cell immunophenotype detected in a cluster of at least 20 cells. The limit of detection (LOD) is 0.001% (1 x 10-5). The assay sensitivity meets or exceeds the 0.01-0.001% (10-4-10-5) level of detection by flow cytometry, as recommended by the NCCN and iwCLL guidelines for MRD analysis in CLL [20,21]. Results between the LOD and LLOQ may be reported as suspicious or equivocal.”

***Comments5: In the previous paper on DNN in B-CLL MRD study (Ref.#8), the Authors have stated that "The hybrid DNN approach...reduced analysis time to approximately 12sec/case" versus 15-20 minutes/case for an experienced technologist. In the present paper the DNN-based "... analysis time was 4.17 ± 2.25 minutes (range: 1.0-10.5 minutes)", which is a remarkable variation. Is this variation taking into account the human-in-the-loop intervention to supervise flagged cases? This point should be adequately discussed.

***Response5: We have now clarified on P3 L98-99 that the “12 sec/case” referred to the time to process 1 millions events per case through the DNN model. It did not include the time to load the file into Infinicyt, perform 2nd review and manual correction, or to save the final file.

SPECIFIC COMMENTS

***Comments6: Just a typo at line n. 52: ' predicotr of survivcal ' instead of 'predictor of survival'.

***Response6: Corrected

***Comments7: Methods: Please specify the meaning of the alpha and beta statistical parameters at line n. 101, for sake of clarity.

***Response7: We have added the following clarification on P4 L119-121: “alpha of 5% representing type I error rate, beta of 20% representing type II error rate or a statistical power of 80%.”

Reviewer 2 Report

Comments and Suggestions for Authors

The authors utilize deep neural networks in processing flow cytometry data of the leukemia patients to identify MRD which is an important clinical task in further patient stratification. The data is thoroughly evaluated statistically and high sensitivity and specificity parameters have been obtained and the time reduction for sample evaluation has been assessed. Such studies are essential for integrating AI into routine patient management safely.

Some points for improvement:

1) This manuscript is missing a well-understandable description of the workflow steps for all the described approaches (classical assessment by technicians, DNN with human in the loop, eventually DNN only) depicting the process from the data acquisition to the diagnosis confirmation. This is only partially addressed in the previous publication (Salama et al 2022 Cancers) and is completely omitted in the present manuscript. One could best describe it with the flow-charts and this will also allow the readers to understand, by automatic DNN processing of which steps such a significant the time efficacy is achieved and which need specialist supervision.

2) Data availability statement does not seem to be rational. If the data was acquired correctly in relation to law and ethics and could be published in a article, there must be no restriction to share the anonymized experimental results.

3) Lines 139-141 It is not completely clear, are the replicates here the same data files copied 4 times or are these different replicate measurements of the same sample?

4) Line 52 - typo "predictor"

Author Response

***Comments1: This manuscript is missing a well-understandable description of the workflow steps for all the described approaches (classical assessment by technicians, DNN with human in the loop, eventually DNN only) depicting the process from the data acquisition to the diagnosis confirmation. This is only partially addressed in the previous publication (Salama et al 2022 Cancers) and is completely omitted in the present manuscript. One could best describe it with the flow-charts and this will also allow the readers to understand, by automatic DNN processing of which steps such a significant the time efficacy is achieved and which need specialist supervision.

***Response1: We have added a new Figure 1 comparing the manual and AI-assisted workflows in Infinicyt. The figure legend explains the opportunity for efficiency gains with the AI-assisted workflow: “Figure 1. Comparison of manual and AI-assisted workflows for flow cytometry analysis in CLL MRD. This flowchart illustrates the key steps in two approaches used to analyze flow cytometry standard (FCS) files for chronic lymphocytic leukemia (CLL) minimal residual disease (MRD) assessment within the Infinicyt software environment. In the manual workflow, FCS files are imported into Infinicyt, gated manually by the user, and saved after interpretation. In the AI-assisted workflow, FCS files are also imported into Infinicyt, where a deep neural network (DNN) is launched via a scripting interface to perform automatic gating; the user then reviews and corrects the results as needed before saving the final analyzed file. Both workflows operate within Infinicyt but differ in the level of automation and required user input. The AI-assisted method enables more efficient case processing while maintaining human oversight.”

***Comments2: Data availability statement does not seem to be rational. If the data was acquired correctly in relation to law and ethics and could be published in a article, there must be no restriction to share the anonymized experimental results.

***Response2: We have corrected the data availability statement as follows: “Raw data were generated at the Mayo Clinic Cell Kinetics Laboratory. Derived data supporting the findings of this study are available from the corresponding author upon reasonable request.”

***Comments3: Lines 139-141 It is not completely clear, are the replicates here the same data files copied 4 times or are these different replicate measurements of the same sample?

***Response3: Four copies of each data files with different file names were created as mentioned here: “Each file was replicated four times.”

***Comments4: Line 52 - typo "predictor"

***Response4: Corrected.

Reviewer 3 Report

Comments and Suggestions for Authors

The manuscript titled “Clinical Validation and Post-implementation Performance Monitoring of a Neural Network-Assisted Approach for Detecting Chronic Lymphocytic Leukemia Minimal Residual Disease by Flow Cytometry” by Seheult et al. represents clinical validation and post-implementation audit of a deep-neural-network (DNN)-based, human-in-the-loop workflow for detecting minimal residual disease (MRD) in chronic lymphocytic leukemia (CLL) by applying multi-color flow cytometry. The manuscript used a large cohort of 240 specimen and applied four different techniques (daily QC, drift detection, error analysis, acceptance sampling) for analytical sensitivity detection which indeed helped in addressing genuine bottleneck of applying AI in diagnostic hematopathology. But I think with better clarification of some concerns along with some minor editorial changes in the manuscript would be beneficial for the comprehensive understanding of the broader audience. In the attached doc file I’ve provided detailed feedback on the manuscript’s strengths and areas of improvement.

Author Response

***Comments1: Although this manuscript is a follow up study of a previously published paper, a concise description of the cohort and training dataset composition should be provided by the authors in the ‘materials & methods’ section for the readers’ convenience.

***Response1: We have rewritten the last paragraph of the introduction to summarize our previous paper on the development of the DNN model, as follows on P2-3 L88-103:

“Our group previously developed a DNN-assisted approach for automated CLL MRD detection using a 10-color flow cytometry panel from 202 consecutive CLL patients status post-therapy collected between February 2020 and May 2021 [8]. This development cohort comprised 143 CLL MRD-positive and 60 CLL MRD-negative samples, with more than 256 million total events analyzed in an 80:10:10% ratio for training, validation, and testing, respectively. The DNN was trained on uncompensated data from a single-tube panel consisting of CD5-BV480, CD19-PE-Cy7, CD20-APC-H7, CD22-APC, CD38-APC-R700, CD43-BV605, CD45-PerCP-Cy5.5, CD200-BV421, Kappa-FITC, and Lambda-PE. This initial work demonstrated high accuracy in detecting CLL MRD and significant workflow improvements, with an average DNN inference time of 12 seconds per case. In the current study, we describe the comprehensive clinical laboratory validation and post-implementation performance monitoring of the DNN-assisted human-in-the-loop workflow for CLL MRD detection by flow cytometry, ensuring that the benefits of AI integration are realized without compromising patient care.”

***Comments2: All the data was originated from 2 Mayo Clinic cytometer. External validation with another laboratory’s dataset and as well as cytometer would help the authors to strengthen the robustness of their claims.

***Response2: The DNN-approach is part of a laboratory developed test being used at a single site. We do not make claims about the generalizability of the model that we have developed or external validation. We expect that each site would need to train and validate an AI model with their local data. We have specifically addressed this limitation on P16 L597-593:

“The limitations of our study include its focus on a specific disease entity (CLL) and flow cytometry panel at a single institution, which limits the generalizability of our findings to other applications and laboratories. Due to the inherent pre-analytical and analytical variability associated with instrument-reagent combinations and panel design, we expect that each laboratory would need to train and validate their own AI model with local data to account for site-specific variables including instruments, reagents, and pre-analytic factors.”

***Comments3: The authors are requested to discuss the false-positive workload and additional review time due to that in more honest way because constant requirement of expert oversight might affect the sustainability of this method. Also, specific strategy should be mentioned to discuss the immunophenotypic features of the missed cases and may be additional data integration would help to improve the model’s sensitivity.

***Response3: We have addressed these concerns in the Discussion on P15 L528-48:

“The error analysis monitoring revealed specific patterns in the performance of the DNN-only, with a high sensitivity (97%) but more modest specificity (59%) when compared to expert review, which is expected since the DNN was optimized for high negative predictive value. It is important to clarify that this specificity concern primarily relates to the DNN-only analysis rather than the DNN+2nd review approach, which demonstrated excellent performance in clinical validation (97.5% accuracy, 100% sensitivity, 94.6% specificity). Our approach intentionally prioritizes high sensitivity and NPV, as false-positive classifications can be readily corrected during expert review, whereas false-negatives are extremely difficult to detect among millions of events. To improve specificity while maintaining sensitivity, we are exploring several strategies including gradient boosting methods, uncertainty quantification techniques, class weightings, and focal loss functions. It is also possible that the incorporation of other markers such as CD79b or T cell markers could improve the performance of the model. The analysis of missed cases identified specific immunophenotypic features that challenge the DNN's classification abilities, particularly CD5-negative B-cell neoplasms (5/13) and lymphoproliferative disorders with atypical immunophenotypes (7/13), with only one classic CLL case missed due to low event counts just below our threshold. This information is valuable for both targeted education of laboratory staff and potential future refinements of the DNN model to improve its performance on these challenging edge cases. Similar error pattern analyses have been proposed to maintain patient safety at the forefront of AI development and facilitate safe clinical deployment [25].”

***Comments4: It would be interesting to add an additional section in the ‘Discussion’ where the authors should mention the clinical impact of the integration of the DNN-assisted workflow in the clinical laboratory, for example if the quicker diagnosis results in better treatment decision or ultimate patient outcome etc.

***Response4: Thank you for your suggestion regarding clinical impact discussion. While this is an interesting consideration, we have intentionally limited our claims to laboratory process improvements rather than clinical outcomes. The primary goal of this study was validating the DNN approach for improving laboratory workflow efficiency and reducing analysis time, which we demonstrated with a 60.3% reduction. Potential downstream clinical benefits—such as faster result delivery, improved inter-rater reliability, or impact on treatment decisions—would require separate dedicated studies with different methodological approaches and clinical endpoints that were beyond the scope of our current validation study. We believe it would be premature to make claims about clinical impact without properly designed studies and corresponding data on human performance without AI to assess and compare these outcomes, though this validation represents an important foundation for such future research.

***Comments5: After a few months of the first publication of this group, another similar paper got published where they used an AI-based commercially available software DeepFlow for detecting MRD in CLL by using 8/10 color flow cytometric detection (DOI: 10.1002/cyto.b.22116). I think this reference should be included in the manuscript and the authors are requested to clarify the advantages and disadvantages of using this kind of commercial software over their DNN-assisted method in the ‘Discussion’ section.

***Response5: Thank you for pointing out the DeepFlow paper. We have included it in our Discussion on P16 L564-586:

“Bazinet et al. recently published their experience using DeepFlow, a commercially available AI-assisted MFC analysis software for CLL MRD detection. They reported overall concordance of 96% between their AI-assisted analysis and expert manual gating, but a lower correlation for quantitative results of 0.865 compared to our workflow [30]. Interestingly, they encountered similar challenges in cases with atypical immunophenotypes, particularly CLL with trisomy 12, which showed higher ex-pression of B cell markers and lower CD43 expression that made AI classification more difficult. While the DeepFlow software offers a user-friendly graphical interface and rapid processing speed (3 seconds for ~1 million events), our DNN-assisted approach offers several advantages. Our model was specifically designed for implementation within existing clinical laboratory workflows, with complete integration into the Infinicyt software platform widely used in flow cytometry laboratories. Additionally, our comprehensive post-implementation monitoring system—including daily electronic quality control, input data drift detection, error analysis monitoring, and attribute acceptance sampling—provides ongoing safeguards that are essential for maintaining reliable performance in routine clinical use. Both studies highlight the potential of AI to dramatically improve laboratory efficiency while maintaining diagnostic accuracy, with each approach offering different advantages for implementation based on a laboratory's specific needs and existing infrastructure. The human-in-the-loop design of both workflows represents a balanced approach to AI integration in clinical laboratories, leveraging the efficiency and standardization benefits of automation while maintaining expert oversight for cases that may challenge the AI model. This aligns with emerging best practices for clinical AI applications, which recognize that human expertise and AI capabilities are often complementary rather than competitive.”

Addressing the above concerns would make the manuscript suitable for publication in this journal.

Reviewer 4 Report

Comments and Suggestions for Authors

Monitoring of measurable residual disease (MRD) in chronic lymphocytic leukemia (CLL) has become a key tool for evaluating treatment efficacy. Despite the use of PCR or NGS techniques, flow cytometry (FCM) remains, according to the ERIC (European Research Initiative on CLL) guidelines, the standard procedure with a detection sensitivity down to 10⁻⁴ (0.01%), and is based on a standardized antibody panel (CD19, CD5, CD20, CD43, CD79b).

The DNN-assisted human-in-the-loop approach to MRD detection in CLL combines artificial intelligence (AI), particularly deep neural networks (DNNs), with oversight by clinical experts. This approach is currently one of the most promising directions in hematologic oncology. Therefore, the choice of topic for this study is very compelling. Nevertheless, researchers still face major challenges such as the ongoing necessity of expert supervision (human-in-the-loop remains essential), and the need to validate AI models in terms of safety and efficacy.

The authors applied FCM using antibodies targeting CD5, CD19, CD20, CD22, CD38, CD43, CD45, CD200, and light chains (kappa or lambda). They demonstrated that the method comparison study showed excellent concordance between the DNN + second-review approach and the manual reference method based on expert analysis, achieving 97.5% overall accuracy for qualitative assessment and a correlation coefficient of 0.99 for quantitative evaluation. These results indicate that the DNN-assisted approach provides essentially equivalent diagnostic information compared to the current reference method, while significantly improving laboratory workflow.

The authors identified limitations of the AI model—a small proportion of cases (2.97%) that fall outside the expected data distribution, primarily high-burden CLL and non-CLL neoplasms with distinct immunophenotypes. This finding emphasizes the importance of understanding the limitations of AI models trained on specific disease entities.

Immunophenotypically, typical CLL is a disease that is relatively straightforward to diagnose using FCM. However, treatment-induced antigenic changes and atypical CLL cases pose a diagnostic challenge—this is something the authors should have mentioned, especially since they are studying MRD in the context of CLL treatment.

Nonetheless, the authors rightly point out that for DNNs, diagnosing CD5-negative lymphoproliferative disorders and non-CLL neoplasms remains a challenge.

The manuscript is well written and, aside from minor comments, can be accepted for publication.

Author Response

***Comments1: The authors identified limitations of the AI model—a small proportion of cases (2.97%) that fall outside the expected data distribution, primarily high-burden CLL and non-CLL neoplasms with distinct immunophenotypes. This finding emphasizes the importance of understanding the limitations of AI models trained on specific disease entities.

***Response1: We agree with the reviewer on the importance of understanding AI limitations especially during clinical validation.

***Comments2: Immunophenotypically, typical CLL is a disease that is relatively straightforward to diagnose using FCM. However, treatment-induced antigenic changes and atypical CLL cases pose a diagnostic challenge—this is something the authors should have mentioned, especially since they are studying MRD in the context of CLL treatment.

***Response2: We have addressed this suggestion in the Discussion on P16 L597-603:

“Our DNN model was trained on post-treatment samples, enabling effective recognition of common treatment-induced immunophenotypic changes such as CD20 negativity and dim CD23 expression. Error analysis revealed that missed cases were primarily attributable to atypical baseline immunophenotypes, rather than treatment-related changes. While our current model handles typical post-treatment modifications well, emerging therapies such as CAR T-cell therapy could introduce new challenges for flow cytometric CLL MRD detection.”

***Comments3: Nonetheless, the authors rightly point out that for DNNs, diagnosing CD5-negative lymphoproliferative disorders and non-CLL neoplasms remains a challenge.

The manuscript is well written and, aside from minor comments, can be accepted for publication.

***Response3: Thank you for the positive feedback.

Round 2

Reviewer 2 Report

Comments and Suggestions for Authors

The authors have addressed all questions.